# Preparation of Large-Size, Superparamagnetic, and Highly Magnetic Fe_3_O_4_@PDA Core–Shell Submicrosphere-Supported Nano-Palladium Catalyst and Its Application to Aldehyde Preparation through Oxidative Dehydrogenation of Benzyl Alcohols

**DOI:** 10.3390/molecules24091730

**Published:** 2019-05-03

**Authors:** Haichang Guo, Renhua Zheng, Huajiang Jiang, Zhenyuan Xu, Aibao Xia

**Affiliations:** 1Zhejiang Key Laboratory of Green Pesticides and Cleaner Production Technology, Catalytic Hydrogenation Research Center, Zhejiang University of Technology, Hangzhou 310014, China; hc.g@163.com; 2School of Pharmaceutical and Material Engineering, Taizhou University, Taizhou 318000, China; zhengrh@tzc.edu.cn (R.Z.); jhj@tzc.edu.cn (H.J.)

**Keywords:** catalytic dehydrogenation, core–shell structure, dopamine, Fe_3_O_4_, nano-palladium

## Abstract

Large-size, superparamagnetic, and highly magnetic Fe_3_O_4_@PDA core–shell submicrosphere-supported nano-palladium catalysts were prepared in this study. Dopamine was encapsulated on the surface of Fe_3_O_4_ particles via self-polymerization and then protonated to positively charge the microspheres. PdCl_4_^2−^ was dispersed on the surface of the microspheres by positive and negative charge attraction and then reduced to nano-palladium. With air as oxidant, the catalyst can successfully catalyze the dehydrogenation of benzyl alcohols to produce the corresponding aldehydes at 120 °C.

## 1. Introduction

Fe_3_O_4_ nanoparticles have been used extensively in many fields due to their superior magnetic properties [1,2,3,4,5] and microwave absorption function [6,7,8,9]. Compared with other metal nanoparticles, the preparation of Fe_3_O_4_ nanoparticles is simple, and the materials used are inexpensive and easily available. Thus, these materials are suitable for industrial production. However, single Fe_3_O_4_ nanospheres will usually experience an agglomeration phenomenon, which hinders the catalytic reaction. Hence, substances such as SiO_2_ [10,11], dopamine [12,13,14,15], and surfactant [16,17,18] are usually used to wrap Fe_3_O_4_ so that it becomes uniformly dispersed, achieving favorable hydrophilic interactions. This type of Fe_3_O_4_ with a core–shell structure has enormous application potential in various fields, including bioseparation [19,20], diagnostic analysis [21,22], molecular imprinting [23], and battery energy storage [24]. Moreover, other metal nanoparticles can be introduced on the shell surface, which not only maintains the original high efficiency of other metal catalysts but also has the magnetism of Fe_3_O_4_. These properties facilitate the separation of catalysts from the reaction mixture.

Benzaldehyde derivative, an important organic chemical raw material, has been widely applied in pesticides [25], medicines [26], spices [27], dyestuff [28], and cosmetics [29]. The preparation of benzaldehyde derivative through catalytic oxidative dehydrogenation is a green synthetic method [30,31]. Magnetic catalysts play a very important role, such as Pd/Fe_3_O_4_ catalysts, which have been successfully studied. In their catalytic reactions, there is a medium yield of benzaldehyde derivatization under oxygen or air [32], which can be greatly improved by adding equivalent molar alkali as co-catalyst [33,34,35]. Moreover, a high yield was achieved when peroxide was used as oxidant [36,37,38]. Therefore, we focused on the development of a new and efficient Fe_3_O_4_@PDA@Pd microparticles catalyst for oxidative dehydrogenation of benzyl alcohols without base under air.

To form the Fe_3_O_4_@PDA core–shell structure, a polydopamine (PDA) layer was wrapped on the surface of large Fe_3_O_4_ particles through dopamine autoagglutination. The amino group of the microspheres was combined with protons through protonation with positive electricity. PdCl_4_^2−^ was dispersed on the surface through positive and negative charge attraction. The large-size, superparamagnetic, and highly magnetic Fe_3_O_4_@PDA core–shell-submicrosphere-loaded nano-palladium catalyst was prepared through further reduction (Scheme 1). This catalyst was applied in the preparation of aldehyde through oxidative dehydrogenation of benzyl alcohol compounds under air.

## 2. Results and Discussion

We characterized and analyzed Fe_3_O_4_@PDA@Pd. As shown in the x-ray diffraction patterns (see Appendix A), the peaks of Fe_3_O_4_ and Fe_3_O_4_@PDA@Pd microparticles were basically identical, that is, they were both characteristic peaks of Fe_3_O_4_. Fe_3_O_4_@PDA@Pd had no characteristic palladium peak, because the palladium particles were very fine under uniform dispersion.

According to the scanning electron microscope (SEM) graph (Figure 1), the particle size of Fe_3_O_4_@PDA@Pd submicrospheres was within 600–800 nm. Thus, their particle size was large, and nano-palladium particles could be seen in the 100,000× SEM graph. During energy-dispersive x-ray spectroscopy (EDS) analysis (Figure 2 and Table 1), the content of palladium weight was 4.24%. In the mapping graph, palladium was uniformly distributed on the surface. According to the mixed weight in the reaction, the theoretical palladium content in Fe_3_O_4_@PDA@Pd was approximately 2.5%, and the palladium content was measured as 2.3% through inductively coupled plasma atomic emission spectroscopy (ICP-AES). Through EDS analysis, the palladium content was 4.24%, as this analysis is mainly aimed at element composition on the material surface, and palladium particles were distributed on the surface of submicrospheres. Moreover, the measured palladium content was partially high.

The transmission electron microscopy (TEM) graph (Figure 3) shows that the prepared Fe_3_O_4_@PDA@Pd catalyst presented a core–shell structure that centered on Fe_3_O_4_ submicrospheres. The dopamine layer was wrapped on the surface of Fe_3_O_4_ submicrospheres, and the thickness of the shell-layer dopamine was uniformly distributed within the interval of 80–90 nm. Nano-palladium particles were dispersed on dopamine. The palladium (0) particle size was within 7–12 nm, and the average particle size was 9.2 nm.

Figure 4 shows the magnetization curves of Fe_3_O_4_ and Fe_3_O_4_@PDA@Pd submicrospheres under room temperature (300 K). As shown in the figure, the maximum saturated magnetic intensity of the two submicrospheres was 75 and 45 emu/g, respectively, and the coercive force for both was 0. The existence of the PDA layer decreased the maximum saturated magnetic intensity of Fe_3_O_4_@PDA@Pd submicrospheres, but they had superparamagnetic and highly magnetic features. Thus, they could easily disperse and separate from the reaction system.

We used Fe_3_O_4_@PDA@Pd for aldehyde preparation through oxidative dehydrogenation of benzyl alcohols. The reaction temperature, reaction time, and oxidizing agent for aldehyde preparation through the Fe_3_O_4_@PDA@Pd-catalyzed oxidative dehydrogenation of benzyl alcohols were optimized using benzyl alcohol as the substrate (Table 2). The results show that the dehydrogenation reaction did not occur when the reaction temperature was low. The increase in reaction temperature promoted a dehydrogenation reaction. However, when the temperature reached 140 °C, benzyl alcohol products were totally converted, and their selectivity rapidly declined. The product was mainly benzoic acid; hence, the appropriate reaction temperature was 120 °C. Similarly, when the reaction time was too long, some products were converted into benzoic acids; thus, the appropriate reaction time was 24 h. When oxygen was used as an oxidizing agent, the conversion rate of benzyl alcohols slightly increased, but selectivity declined, indicating that benzaldehyde was easily oxidized under oxygen conditions. Partial benzyl alcohol was converted into benzaldehyde when no oxidizing agent was used, indicating that this catalyst could catalyze benzyl alcohol dehydrogenation even under oxygen-free conditions so as to prepare benzaldehyde. However, the conversion rate was low.

With optimal reaction conditions (Table 2, Entry 4), we checked the reaction scope, and the results are shown in Table 3. There was a medium yield of synthetic aldehyde prepared through Fe_3_O_4_@PDA@Pd-catalyzed substituent benzyl alcohol dehydrogenation. The dehydrogenation of substituent benzyl alcohols was easier under high-electron cloud density (Entries 2, 3, 4, 8) than under low-electron cloud density (Entries 5, 6, 7). When a substituent group (Entries 1, 9) was present at the adjacent position, it could exert an inhibitory effect on the dehydrogenation reaction of benzyl alcohols, possibly because the existence of adjacent groups affected C–OH bond adsorption by the catalyst. In addition, heterocyclic ring aromatic methanol (Entries 11, 12, 13, 14) could also experience oxidative dehydrogenation to generate the corresponding aldehyde, and a large quantity of acids was generated after 24 h reaction of 2-furancarbinol (Entry 11) and 2-thienylmethanol. All products were characterized by ^1^H NMR and ^13^C NMR (see Appendix A).

Fe_3_O_4_@PDA@Pd permits easy recovery of the catalyst from the reaction mixture by a magnet. The reusability of the recovered catalyst was evaluated for the oxidative dehydrogenation of benzyl alcohol as a model reaction, and nearly 95% of its original activity was retained even after five cycles (Figure 5). After six cycles, its activity decreased but its catalytic selectivity remained almost invariant. Compared with the fresh catalysts by SEM and EDS (see Appendix A), we found that the catalysts were partially fractured and the palladium content was reduced by nearly half.

## 3. Materials and Methods

Commercially available reagents were used without further purification. The x-ray diffraction (XRD) analysis used a Bruker AXS D8 Advance diffractometer. Scanning electron microscope (SEM) and energy-dispersive spectrometry (EDS) images were recorded on a Hitachi S-4800 field emission scanning electron microscope equipped with an energy-dispersive spectrometer. Transmission electron microscopy (TEM) images were obtained on a JEOL JEM-2100F field transmission electron microscope. Magnetic information was obtained on a Quantum Design DynaCool-9 vibrating sample magnetometer. The ^1^H NMR and ^13^C NMR spectra were recorded on a Bruker Avance 400 MHz spectrometer using tetramethylsilane (TMS) as internal standard.

Fe_3_O_4_ particles were prepared according to the method specified in Reference [39]. 

The general procedure was for the preparation of Fe_3_O_4_@PDA. Approximately 0.15 g of strong ammonia water was dissolved in 50 mL of deionized water. Then, 0.1 g of Fe_3_O_4_ and 0.16 g of dopamine hydrochloride were added, and the mixture was placed under uniform ultrasonic dispersion and mechanical agitatation under 40 °C for 24 h. After the reaction ended, solid and liquor were separated using a magnet. The product was placed under ultrasonic washing using 25 mL × 3 deionized water and 25 mL × 3 ethanol, and the solid was directly used in the next step.

The general procedure for the preparation of Fe_3_O_4_@PDA@Pd was as follows. To prepare palladium sodium chloride solution, 0.115 g (approximately 0.65 mmol) of palladium chloride and 0.076 g (approximately 1.3 mmol) of sodium chloride were taken, and deionized water was added dropwise until the weight reached 8 g. The mixture was heated to 50 °C, and a palladium sodium chloride solution was prepared and placed in a brown bottle for later use. To prepare Fe_3_O_4_@PDA-loaded nano-palladium, Fe_3_O_4_@PDA solid was placed under ultrasonic washing using 25 mL of deionized water, 0.1 N of 25 mL hydrochloric acid, 25 mL of deionized water, and 25 mL of ethanol. Approximately 40 mL of ethanol and 4 mL of deionized water were added, and the mixture was mechanically agitated under 10 °C. Subsequently, 0.29 g of palladium sodium chloride solution was slowly added dropwise, and then the mixture was continuously agitated for 3 h. A solution comprising 60 mg of ascorbic acid and 6 mL of deionized water was added slowly using a needle cylinder for 20 min. Afterward, the reaction lasted for 2 h. The solid and liquor were separated using a magnet; the reaction product was placed under ultrasonic washing using 25 mL of ethanol, 25 mL × 3 deionized water, and 25 mL × 3 ethanol. The solid was preserved in 25 mL of ethanol and sealed using nitrogen (solid was approximately 0.1 g after drying).

The general procedure for the oxidative dehydrogenation of benzyl alcohol was as follows. Approximately 0.1 g of Fe_3_O_4_@PDA@Pd catalyst (2 mol% of Pd) was used. The aforementioned solid and liquor were separated using a magnet and placed under ultrasonic washing using 10 mL × 3 O-xylene. Then, 1 mmol of benzyl alcohol derivative and 5 mL of O-xylene were successively added and blended under ultrasonic conditions. The mixture was then magnetically stirred for 24 h under 120 °C. After the reaction ended, the solid and liquor were separated using a magnet. The liquor was concentrated, and the dehydrogenation product could be obtained after purification through column chromatography (petroleum ether/ethyl acetate).

## 4. Conclusions

We developed a method for preparing large-size, superparamagnetic, and highly magnetic Fe_3_O_4_@PDA core–shell submicrosphere-supported nano-palladium catalysts. The catalysts could catalyze the dehydrogenation of benzyl alcohols to produce the corresponding aldehydes with medium to high yields under air as oxidant without base. Additionally, they could be easily removed from the reaction media by an external magnet and reused five times without a considerable reduction in reactivity.

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
