# Peer review of "Preparation of Large-Size, Superparamagnetic, and Highly Magnetic Fe3O4@PDA Core–Shell Submicrosphere-Supported Nano-Palladium Catalyst and Its Application to Aldehyde Preparation through Oxidative Dehydrogenation of Benzyl Alcohols"

_molecules, 2019, doi:10.3390/molecules24091730_

Round 1

Reviewer 1 Report

This paper describes a preparation of Fe3O4@PDA@Pd and its application to aldehyde preparation. The catalyst could be fully characterized by ICP-AES, XRD, EDS and TEM. However, efficient method using Fe3O4 has been developed for dehydrogenation of benzyl alcohols (see, Sarma et al. ACS Omega 2018, 3, 13711−13719). Therefore, I think this manuscript does not meet the novelty and impact requirements of this journal. I suggest publishing this material in a more specialized journal. The following points should be considered before the publication. Page 2: Proposed mechanism for dehydrogenation of benzyl alcohols should be added (for more details, roles of Fe and Pd, O2 etc.).

Author Response

Dear referee:

Thanks for your kindly comments. We have revised our manuscript and our replies of your comments are as bellow:

Point 1: About the mechanism.

Response 1: As the detailed mechanism is not clear enough, so a logical proposed mechanism was present here (see our notes file).

Thank you very much for your attention and consideration. If you have any further questions about our revised manuscript, please don’t hesitate to let me know.

Yours sincerely,

Aibao Xia (Associate Professor), Zhenyuan Xu (Professor)

Catalytic Hydrogenation Research Centre,

State Key Laboratory Breeding Base of Green Chemistry-Synthesis Technology,

Zhejiang University of Technology,

Hangzhou, 310014, China

Tel/Fax +86 (571) 88320066, 

E-mail: xiaaibao@zjut.edu.cn, greenchem@zjut.edu.cn.

Reviewer 2 Report

In this work, the oxidative and non-oxidative dehydrogenation of benzyl alcohol and substituted benzyl alcohols has been carried out over highly magnetic Fe3O4 with dopamine encapsulated on the surface and also with PdCl42- dispersed on the surface of the Fe3O4 microspheres.

Please find some comments/suggestions below for improving the manuscript:

 1. The authors should make a characterization of the catalyst after the reaction and compare with the fresh catalyst.

2. It is necessary to compare the results with literature data.

3. The authors do not specify how they made the product identification for dehydrogenation.

4. Conclusions should be improved.

Author Response

Dear referee:

Thanks for your kindly comments. We have revised our manuscript and our replies of your comments are as bellow:

Point 1: The authors should make a characterization of the catalyst after the reaction and compare with the fresh catalyst.

Response 1: The characterization by SEM and EDS of the catalyst after the reaction was performed. And some discussions were added in line 129-135 of page 5 of the text.

Point 2: It is necessary to compare the results with literature data.

Response 2: Some discussions were added in line 41-47 of page 1-2 of the text.

Point 3: The authors do not specify how they made the product identification for dehydrogenation

Response 3: The Supplementary Materials was attached.

Point 4: Conclusions should be improved.

Response 4: The conclusion was rewritten in page 7 of the text.

Thank you very much for your attention and consideration. If you have any further questions about our revised manuscript, please don’t hesitate to let me know.

Yours sincerely,

Aibao Xia (Associate Professor), Zhenyuan Xu (Professor)

Catalytic Hydrogenation Research Centre,

State Key Laboratory Breeding Base of Green Chemistry-Synthesis Technology,

Zhejiang University of Technology,

Hangzhou, 310014, China

Tel/Fax +86 (571) 88320066, 

E-mail: xiaaibao@zjut.edu.cn, greenchem@zjut.edu.cn.

Reviewer 3 Report

Xia, Xu, and coworkers presented in this manuscript the preparation of Fe3O4@PDA@Pd and its application in alcohol oxidation to the produce aldehyde. After the consideration of the following points, we recommend a major revision of the this manuscript.

1.      No recycling test was reported. At least, the authors should mention their attempts.

2.      No cross-comparison of the catalytic activities of their catalytic system with those published in the literature.

3.      Literature on Pd/Fe3O4 were missing.  

4.      The Pd loading (mol% or g%) for the catalytic reaction was not stated.

5.      Since no characteristic peaks of Pd was observed in the XRD pattern in Fig. 1, the figure should better moved to the supporting information.

Author Response

Dear referee:

Thanks for your kindly comments. We have revised our manuscript and our replies of your comments are as bellow:

Point 1: No recycling test was reported. At least, the authors should mention their attempts.

Response 1: The recycling test and some discussions were added in page 5-6 of the text. The catalyst original activity was retained even after five cycles.

Point 2: No cross-comparison of the catalytic activities of their catalytic system with those published in the literature.

Response 2: In the dehydrogenation of benzyl alcohols to produce corresponding aldehydes, the catalysts, Fe3O4@PDA@Pd, can achieve similar catalytic activities with those published in the literature. And the catalysts can perform well even under air as oxidant without base.

Point 3: Literature on Pd/Fe3O4 were missing.

Response 3: References were added as references 32-38.

Point 4: The Pd loading (mol% or g%) for the catalytic reaction was not stated.

Response 4: The Pd loading is 2 mol% for the catalytic reaction as line 171 of page 6 of the text.

Point 5: Since no characteristic peaks of Pd was observed in the XRD pattern in Fig. 1, the figure should better moved to the supporting information.

Response 5: The Figure was moved to the Supplementary Materials.

Thank you very much for your attention and consideration. If you have any further questions about our revised manuscript, please don’t hesitate to let me know.

Yours sincerely,

Aibao Xia (Associate Professor), Zhenyuan Xu (Professor)

Catalytic Hydrogenation Research Centre,

State Key Laboratory Breeding Base of Green Chemistry-Synthesis Technology,

Zhejiang University of Technology,

Hangzhou, 310014, China

Tel/Fax +86 (571) 88320066, 

E-mail: xiaaibao@zjut.edu.cn, greenchem@zjut.edu.cn.

Round 2

Reviewer 1 Report

I would recommend publication of the revised manuscript in Molecules.

Author Response

Dear referee:

Thanks for your kindly comments. We have revised our supplementary notes. 

Thank you very much for your attention and consideration. If you have any further questions about our revised manuscript, please don’t hesitate to let me know.

Yours sincerely,

Aibao Xia (Associate Professor), Zhenyuan Xu (Professor)

Catalytic Hydrogenation Research Centre,

State Key Laboratory Breeding Base of Green Chemistry-Synthesis Technology,

Zhejiang University of Technology,

Hangzhou, 310014, China

Tel/Fax +86 (571) 88320066, 

E-mail: xiaaibao@zjut.edu.cn, greenchem@zjut.edu.cn.

Reviewer 2 Report

Agree with publication in this form

Author Response

(The authors gave the same response as above.)

Reviewer 3 Report

This manuscript has been revised appropriately. I can recommend publication of this manuscript. One minor thing is that the catalytic products are known compound. Their NMR assignments should be supported by literature references.  

Author Response

Dear referee:

Thanks for your kindly comments. We have revised our supplementary notes and our replies of your comments are as bellow:

Point 1: Their NMR assignments should be supported by literature references.  

Response 1: The NMR assignments of the catalytic products have been supported by literature references in the supplementary notes.

Thank you very much for your attention and consideration. If you have any further questions about our revised manuscript, please don’t hesitate to let me know.

Yours sincerely,

Aibao Xia (Associate Professor), Zhenyuan Xu (Professor)

Catalytic Hydrogenation Research Centre,

State Key Laboratory Breeding Base of Green Chemistry-Synthesis Technology,

Zhejiang University of Technology,

Hangzhou, 310014, China

Tel/Fax +86 (571) 88320066, 

E-mail: xiaaibao@zjut.edu.cn, greenchem@zjut.edu.cn.
